# Psychiatric symptoms and emotion regulation strategies among the unemployed people in Korea: A latent profile analysis

**Min Sun Kim**[ID]*

Department of Psychology and Psychotherapy, Dankook University, Dongnam-gu, Cheonan-si, Chungnam, South Korea

* kms82qwert@hanmail.net

**Data Availability Statement:** All relevant data are available from Dryad (DOI: 10.5061/dryad.3n5tb2rds).

**Funding:** The author(s) received no specific funding for this work.

## Abstract

This study aimed to explore the profiles of emotion regulation strategies among unemployed people, and to examine the association of latent profiles with demographics and psychiatric symptoms. The study included 136 men (42.8%) and 182 women (57.2%). The average age of the participants was 35.84 years (SD = 26.83). Latent profile analysis was used to determine emotion regulation strategy profiles. Associated factors of profile membership were identified with multinomial logistic regression. The four-profile model (low adaptive emotion regulation class, low negative emotion regulation/moderate positive regulation class, high negative emotion regulation/support-seeking class, adaptive emotion regulation class) was selected as the best solution. As a result of examining the probability of being classified into each class according to emotional difficulties, the lower the level of anxiety and somatization, the higher the probability of belonging to the class 2 adaptive emotion regulation class (n = 56, 18%). The higher the depression, the higher the probability of being classified into class 4 (n = 65, 20%) using a lot of negative emotion regulation strategies. The results of this study indicate that unemployed people can be classified into various subgroups according to their emotion regulation strategies. Also, the probability of being classified into each subgroup was different based on the types of emotional difficulties such as depression, anxiety, and somatization. Through the results of this study, it is possible to understand the relationship between the psychiatric symptoms of unemployed people and emotion regulation strategies and to suggest methods for promoting effective emotion regulation strategies among this population group.

## Introduction

Korea has been in a spiral of unprecedented unemployment since it received bailouts from the International Monetary Fund (IMF). In mid 1998, when restructuring, including layoffs were underway, the official real number reached 1.6 million, raising the sense of social crisis. The IMF's bailout seemed to have solved the problem of unemployment, but in 2001, the number of official unemployed workers exceeded 1 million again. In addition, the first restructuring by

**Competing interests:** The authors have declared that no competing interests exist.

layoffs was mainly focused on manufacturing workers, but after the 1980s, middleclass managers, experts, and office workers were the main targets, and the layoff rate of men and higher education workers has been increasing gradually [1] According to the Organization for Economic Cooperation and Development (OECD) Employment Outlook 2018, Korea had the highest turnover rate among the OECD countries [2] at 31.8% and a relatively weak social safety net for loss of income in the reemployment process. Also, the reemployment rate in Korea within 1 year after the dismissal of a worker was 46.1%, which was slower compared with other OECD countries such as the United States (57.8%), Japan (50.8%), and Australia (73.6%).

In the absence of a social security system, unemployment not only poses a fundamental threat to the right to live for individual workers, but also has a negative effect on the entire family, which can threaten the structure of the family and the overall social structure. The prevalence of family crises among Koreans is 50%, and the frequency of economic crisis (e.g., unemployment) was the highest among the family crises [3]. In periods of economic recession and overall industrial change, problems related to unemployment can be a factor that makes the social cost more burdensome. Therefore, it is necessary to expand mental health programs that can facilitate the process from unemployment to reemployment, and to focus more on the mental health problems of unemployed people at the national level.

From a psychological perspective, unemployment is linked to the following negative results: First, it was found that the loss of economic independence and social role performances due to the decrease of economic resources obtained from a fixed income were insolvent. Second, it leads to involuntary interruption of the mutual relationship between individuals and the organization. Third, unemployment means loss of work, which, for many, is the basis of achievement, self-realization, and self-identity [4, 5, 6]. Also, the increase in anxiety and tension due to unemployment has been demonstrated by many empirical studies [7, 8, 9]. McKee-Ryan, Song, Wanberg, and Kinicki [10] examined 52 cross-sectional studies that compared the health of unemployed and employed people and found that those who were unemployed had significantly poorer mental health, life satisfaction, family satisfaction, and subjective physical health. Unemployment has been linked to increases in the suicide rate among middle-aged people. According to the distribution trend of suicide rates by age in Korea (the number of suicides per 100,000 ages), the suicide rate of young people in their 20s was the highest in 1983, but the suicide rate of young people in their 20s was lowered in 1993, while the suicide rate of older people rose [11]. Comparing the distribution of suicide rates by age in 1993 and 2003, it can be seen that the suicide rate is increasing as the age of the population increases, and the suicide rate of adults aged over 50 years has been increasing very rapidly recently.

In stress situations, unemployed people are more likely to cope with stress by self-reading or wishful thinking rather than actively coping and avoiding behavior. In a study by Ahn, Tak, Yoo, Han, and Han [12], dysfunctional coping, reemployment restriction requirements, employment commitment, and job search intensity had negative effects on mental health, whereas self-esteem, life satisfaction, adaptive coping, and social support had positive effects. However, most of the studies conducted in Korea have been based on specific mediating variables such as family resilience [13] and perceived social support [14, 15]. Therefore, there is a relative lack of understanding of the various emotion regulation strategies and influences used by unemployed people. Therefore, this study examined the types of spontaneous groups that are derived according to the emotional-centered coping strategies for unemployed people, and explored the probability of being classified into each group according to the psychiatric symptoms.

## Emotion regulation strategies

Theories that explain emotion regulation explain the influence of various strategies to control emotions. The most well-known theory, the process model, explains the process of emotion regulation by dividing the emotion before and after the emotional expression based on the flow of time [16]. The focus is mainly on the positive reinterpretation of the situation, such as the cognitive reinterpretation of the situation before the emotional expression. Next, the strategies that are implemented after emotional expression focus on emotion regulation strategies during emotional manifestation, such as emotional suppression, problem-solving, acceptance, and rumination. Although some of the emotion regulation strategies used in the process of controlling emotions are effective at reducing unpleasant emotions, there are also strategies that may cause the development of various psychiatric symptoms and psychopathology through the accumulation and exacerbation of unpleasant emotions [15, 17, 18]. Some researchers have focused on the specific strategies mobilized for emotion regulation and have investigated the relationship between emotion regulation strategies and psychiatric symptoms [19–21]. In Lee JY [18]'s study, negative emotion regulation strategy was shown to increase psychopathology, and only support seeking strategy among positive emotion regulation strategy lowered psychopathology.

This study assumed that various subgroups would be derived according to subfactors of emotion regulation strategies based on findings from previous studies, which are as follows: In a study by Eftekhari, Zoellner, and Vigil [22], the emotional control group was divided into the high emotional control group, high cognitive reinterpretation-low oppression group, medium cognitive reinterpretation-low suppression group, and low emotional control group. The high cognitive reinterpretation-low repression group showed lower levels of depression, anxiety, and post-traumatic stress disorder scores than the other groups, and the low emotional control group showed higher levels of depression, anxiety, and post-traumatic stress disorder. In the study of Lee, Kim, and Choi [23], four groups was derived based on emotional recognition and expression: emotion recognition, shield behaviors to not recognize emotions, negative emotional expression behavior, and regulation behavior. The results of the study showed that emotional difficulties such as somatization, obsession, depression, and anxiety were high in the negative emotional expression behavior group.

Some studies [22–24] have been conducted based on the assumption that there are various subgroups according to the emotion regulation strategy, and the more positive emotion regulation strategies were used, the lower Psychiatric Symptoms was. However, there are limitations to this approach. First, the variable-centered approach does not take into account that unemployed people can use multiple emotion regulation strategies in combination. For example, the variable-centered approach focuses mainly on the linear relationship between variables and ignores the fact that variables can be combined in new ways. The variable-centered approach and person-centered approach are similar in that both only interpret emotion regulation and the relationship between variables. In the variable-centered approach, the effect of the variables is independently verified through individual differences, and in the person-centered approach, the relationship between the combination of variables and different strategies is verified [25]. This study attempts to shed light on the complicated emotional processes of unemployed people by examining various emotion regulation strategies together, unlike the existing variable-centered approach that regards negative emotion regulation strategies and positive emotion regulation strategies as mutually exclusive. Second, some study [26, 27] that have examined subgroups according to emotion regulation strategies have focused on cognitive efforts such as suppression, avoidance, and reinterpretation, and because they mainly used emotion regulation strategies from stress-coping theory, they do not include comprehensive

strategies. Lastly, the studies [22, 23] on the subgroups according to the emotion regulation strategies for the general public are classified by cluster analysis, so the group classification can be somewhat arbitrary and it is limited to generalize to individuals in the stress situation of unemployment.

## Emotion regulation strategies and psychiatric symptoms

Previous studies [28–32] found that the use of specific emotion regulation strategies (e.g., suppression of expression, thought suppression, rumination) and the low use of other strategies (e.g., reinterpretation and self-disclosure) were related to the level of depression symptoms. Studies that have examined emotion regulation strategies of people experiencing depression focused mainly on cognitive emotion regulation, and only on examining the influence of specific emotion regulation strategies such as avoidance [33], rumination [20, 34], and reinterpretation [32] among cognitive emotion regulation. In fact, cognitive emotion regulation processes such as reinterpreting or changing the situation are linked to emotional expressions, responses, and self-disclosure, and there are also people who use various strategies at the same time rather than as a conscious emotion regulation strategies [23, 35].

People with high trait anxiety can be sensitive to stress situations and show excessive anxiety. Caver, Scheier, and Weintraub [36] noted that high trait anxiety tends to avoid stimuli. Therefore, those who feel anxiety due to their personality characteristics are highly concerned about vague threats that have not occurred and try to control their anxious emotions. People with panic disorder, for example, can explain that the "fear by fear" experience contributes to the maintenance and triggering of seizure symptoms [37]. This leads to the use of emotion regulation strategies to avoid or suppress fears because fears are not accepted and negative emotions are formed. Furthermore, anxiety disorder can lead to misinterpretation of general emotional responses and thus dysfunctional emotion regulation attempts are generated. In contrast to the control group with low anxiety levels, the emotional disorder group has low emotional acceptance and awareness levels and the transitional disorder group has low emotional acceptance and awareness levels [28, 38].

Next, this study examined whether the probability of being classified into each subgroup was different according to the somatization level. The correlation between negative emotions and physical symptoms is known to be .30~.50, and the more negative emotions, the more likely a person is to complain about physical symptoms [39–44]. However, there have been reports that somatization problems occur when suppressing emotions rather than with negative emotions [42, 45]. Some studies [46, 47] have noted that people with high somatization may have a high belief in negative emotions, and that if they cannot control unpleasant emotions at a cognitive level, they may show exaggerated physical and behavioral responses in stress situations and become vulnerable to diseases.

However, recent studies have shown that there are some inconsistent results in explaining the relationship between psychopathology, positive emotional regulation strategies, and negative emotional regulation strategies. For example, Schfer, Naumann, Holmes, Tuschen-Caffier, and Samson [48], which meta-analyzed existing studies on the relationship between emotional regulation strategies, depression, and anxiety, have shown that positive emotional regulation strategies such as cognitive reinterpretation have more important effects on psychiatric symptoms such as depression and anxiety than negative emotional regulation strategies such as rumination and avoidance. This means that enhancing positive emotional regulation strategies can have a more positive effect on depression and anxiety of adolescents than reducing negative emotional regulation. On the other hand, Aldao and Nolen [49] 's study showed that the relationship between positive emotion regulation strategies and psychiatric symptoms were

more significant for adults who use high negative emotion regulation strategies. This means that the positive emotion regulation strategies is more effective in lowering the psychopathology of people who use negative emotion regulation and people use negative and positive emotion regulation strategies flexibly according to the situation [8, 50, 51].

## The present study

In this study, we have focused on identifying the characteristics of each subgroup by classifying unemployed people into subgroups according to the use of emotional regulation strategies through latent profile analysis. This study derived latent profiles based on one negative emotion regulation strategy and four adaptive emotion regulation strategies (Fig 1). We also examined whether the probability of belonging to each group derived based on emotion regulation strategies differs according to the psychiatric symptoms of unemployed people. According to previous studies, people with vulnerability to depression were more likely to use negative emotional regulation strategies such as avoidance [33], rumination [20, 33], and reinterpretation [32]. In addition, the group with low emotional acceptance and awareness showed higher anxiety level [28, 38], and the group with low emotional acceptance and awareness showed higher somatization in the group without expressing negative emotions [42, 45]. In this study, the probability of being classified into each group would be different according to psychiatric symptoms of the unemployed people.

Research Question 1: Are there qualitatively distinct emotional regulation strategy profile of unemployment people?

Research Question 2: Do demographic information and psychiatric symptoms predict emotional regulation strategy profile membership?

## Methods

### Participants and procedure

The gender of the participants was 136 males(42.8%) and 182 females(57.2%), and the average age was 42.55(*SD* = 7.40). The educational background was 70(22.0%) high school graduates, 37(11.6) technical college graduates, 182(57.2%) university graduates, and 29(9.1%) graduate students. The retirement types were voluntary retirees(58.2%) and involuntary retirees (41.8%). The number of jobless was 55(17.3%) once, 103(32.4%) twice, and 160(50.3%) three times and more. The average job search period was 35.84(*SD* = 26.83) months.

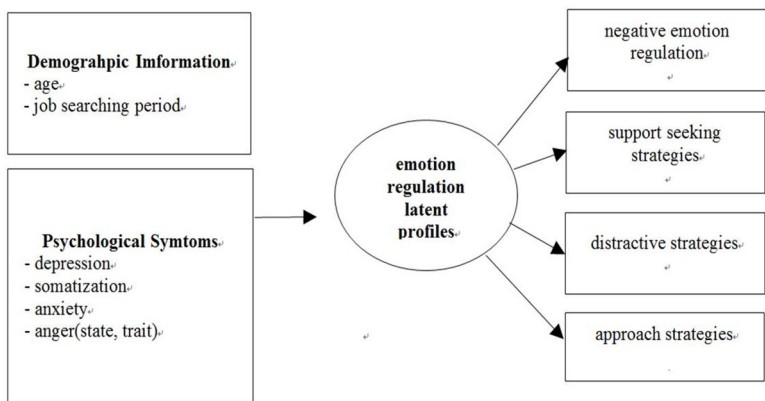

**Fig 1. Research model.**

In order to recruit participants, the purpose of the study was explained to the institutions that help reemployment or job search such as the vocational education center under the Ministry of Labor and the Women's Job Center. In this study, participants were selected from adults over 20 years old who do not currently have a full-time job and want to find a job. The study was conducted in accordance with the Declaration of Helsinki. The survey kits were distributed to the institutions that allowed the participation in the study, and the questionnaires were distributed to the people who were willing to participate in the study. This study explained the necessity and purpose of the study to the participants and received oral consent. The researchers explained that there is no disadvantage due to the absence of research and that they can withdraw if there is no intention to participate in the survey.

## Measures

**Emotional regulation strategy.**   In this study, the emotion regulation strategy questionnaire(ERSQ) developed by Lee and Kwon [52] was used to measure the emotion regulation strategy. ERSQ is divided into cognitive, experiential, and behavioral strategies according to what aspect of emotion is approach ed and emotional changes are caused. A total of 16 strategies including 5 cognitive strategies, 5 experiential strategies, and 6 behavioral strategies are measured. A result of Lee and Kwon [52] 's research, 16 emotion regulation strategies were classified into maladaptive strategies, support-seeking strategies, attention distraction strategies, and approach strategies. Maladaptive strategies showed significant positive correlation with psychopathology such as depression, while other strategies showed negative correlation [53, 54]. Previous research [52] classified the other three strategies as adaptive ones except for maladaptive emotion regulation. Lee and Kwon referred to support-seeking strategies, attraction differentiation strategies, and approach strategies as adaptive strategies. This study classified latent profiles according to emotional regulation strategies of unemployed people based on ERSQ scale, which measures various emotional regulation strategies.

**Depression.**   In this study, Beck Depression Inventory (BDI) [55] was used to measure depression. BDI is a self-reported test developed to measure the severity of symptoms and the presence of depression symptoms. This study used the scale adapted and validated by Lee and Song [52] for Korean people and consists of 21 questions in total. Each item is scored from 0 to 3 and the total score is from 0 to 63. In the study of Lee and Song [56], the overall reliability of BDI for the general people was .78 and .85 for the depressed patients. The reliability of the existing Zung self-rating depression scale (SDS) [57] and MMPI-Depression scale (MMPI-D) [58] was .70 and .50 in the depressed group, respectively.

**Somatization.**   The somatization was used by extracting questions about the somatization symptoms appeal among the nine sub-factors of the scale validated by Kim, Kim, Won [59] in the Korean version of SCL-90(Brief Symptom Inventory & Matching Clinical Rating Scale) developed by Derogatis [60]. The question is designed to measure the 12 symptoms of headache, dizziness, heavy arms and leg et cetera on a 5-point Likert scale. Participants recall their experiences over the past week and respond to how similar they are to each item. The range of total score is continuous variable from 12 to 60, and the higher the score, the more severe the somatization symptoms. The reliability of this scale is presented in each area, and the reliability of the somatization factor at the time of development of the scale .79.

**Trait anxiety.**   In this study, the State-Trait Anxiety Inventory(STAI) which was produced by Spielberger [61] and adapted by Kim [11] was used to measure trait anxiety. The trait anxiety is a trait that is not affected by psychological tension according to the situation as a relatively unchanging individual difference. It consists of 20 items and each item is evaluated as

4-points on the Likert scale. The range of individual scores is from 20 to 80, and the higher the score, the higher the trait anxiety.

## Analysis

This study was conducted to classify latent profiles according to emotional regulation strategies for unemployed people and to verify the influence of related variables. To do this, the Latent Profile Analysis(LPA) was conducted using the Mplus 7.0 program. When classifying groups in LPA, the subgroups estimated through the analysis of the responses are called latent profiles. Unlike the existing variable-centered approach to understand the phenomenon through the relationship of variables, latent profile analysis is based on the person-oriented approach that approaches with interest in individual characteristics [40]. In addition, unlike cluster analysis is that sorts groups somewhat subjectively, LCA sets a model and determines the number of groups through statistical procedures, and it is easy to interpret using probability.

In this study, the suitability of the model was evaluated based on the quality of classification, information index, and model comparison verification for the final model selection. To classify the quality of the model, the Entropy index was used. The Entropy index was between 0–1 and the probability of belonging to one latent class was close to 1, and the value increased as the rate of belonging to another latent class was close to 0. The value of about 0.8 is decided as the good classification [62]. Next, the information index used AIC (Akaike Informaton Criterion) [63], BIC(Bayesian Information Criterion) [64], SABIC (Sample-size Adjusted BIC) indexes, and the lower the value, the better the fit. Finally, the LMRLRT(Lo-Mendell-Rubin adjusted Likelihood Ratio Test) [65] and the BLRT(Parametric Bootstrap Likelihood Ratio Test: BLRT) [66] were used for the model comparison verification. When evaluating the model with the number of latent profiles k, both of the results are used to verify the difference between the k-1 latent profile and the k-1 latent profile model. The p-value of the result is used to evaluate whether k-1 latent profile model are rejected to support the k latent profile model. If the latent profile model of k is not significant, the latent profile model of k-1 is selected. In this study, the influence of independent variables was verified through the polytomous logistic regression presented in the final model analysis results.

## Results

### Descriptive analysis

Correlations and means/standard deviations of all variables are shown in Table 1. The results of correlation analysis showed that negative emotion regulations strategies had a significant positive correlation with depression, somatization, and anxiety. Also, the three types of adaptive emotion regulation strategies were found to have a significant negative correlation with depression and anxiety. On the other hand, attention distractive and approach strategies showed significant negative correlation with somatization and state anxiety, while support seeking strategies did not show significant correlation.

### Identification and description of latent classes

Model fit indices for the five classes are reported in Table 2. As a result of the study, when AIC, BIC, and SABIC compared to 2 class solution, the value became smaller as the number of profiles increased significantly, and the Entropy value was .80 in 4 class solution, which was suitable for classification criteria. Class 5 also meets the criteria for the quality of classification and

**Table 1. Bivariate correlations and descriptive statistics.**

|  | 1 | 2 | 3 | 4 | 5 | 6 | 7 | 8 | 9 |
|---|---|---|---|---|---|---|---|---|---|
| 1. age | - | | | | | | | | |
| 2. year of job seeking | -.06 | - | | | | | | | |
| 3. negative emotion regulation | -.08 | .17** | - | | | | | | |
| 4. attention distractive strategies | .01 | -.08 | .11* | - | | | | | |
| 5. approach strategies | .03 | -.11* | .07 | .71** | - | | | | |
| 6. support seeking strategies | -.07 | -.13* | .23** | .66** | -.62** | - | | | |
| 7. depression | -.06 | .19* | .48** | -.05 | -.07 | -.01 | - | | |
| 8. somatization | -.15** | .10 | .43** | -.41** | -.37** | -.29** | .48** | - | |
| 9. anxiety | -.04 | .10 | .44** | -.38** | -.34** | -.29** | .54** | .77** | - |
| *M* | 42.55 | 35.84 | 2.37 | 3.23 | 3.32 | 3.15 | 2.04 | 2.39 | .73 |
| *SD* | 7.40 | 26.83 | .70 | .74 | .64 | .77 | .83 | .59 | .57 |

*p < .05

**p < .01.

information index, but the group of less than 5% of the total participants is derived and did not consider it as the final model. Shin and Son [44] mentioned that it is difficult to interpret meaningfully in small groups of less than 5% of the total cases, and judged that the group was not suitable for the classification of potential profiles. In this study, the 4 class model was selected as the most suitable model by combining the quality of classification and information index results.

Fig 2 depicts the pattern of mean scores across the latent classes of the four class model. The names of each classes were based on the level of negative emotion regulation and three positive emotion regulations. First, class 1 was named as "low emotion regulation class" (n = 48, 15%) as the group with the lowest use of positive emotion regulation strategies. The second group named the fourth group was named as "adaptive emotional regulation class" (n = 56, 18%) using low negative emotion regulation strategy and high positive strategies. The third group was named "low negative emotional regulation-moderate positive regulation class" (n = 149, 47%) because all the emotion regulation strategies appeared to be intermediate level. Finally, the fourth group was named "high negative emotional regulation-support seeking class" (n = 65, 20%) as a class that used negative emotion regulation the most and used support seeking emotional regulation strategy.

**Table 2. Fit indices for one-to five class models.**

| Model | AIC | BIC | SABIC | Entropy | LMR_LRT | BLRT |
|---|---|---|---|---|---|---|
| 2 class solution | 2413.36 | 2481.07 | 2423.98 | .784 | -1367.27*** | 351.08*** |
| 3 class solution | 2313.87 | 2419.21 | 2330.40 | .776 | -1188.68*** | 117.44 |
| **4 class solution** | **2229.18** | **2372.14** | **2251.61** | **.803** | **-1128.94**** | **102.90**** |
| 5 class solution | 2207.83 | 2288.41 | 2236.17 | .804 | -1076.59 | 40.65 |
| | 2147.31 | 2365.50 | 2181.54 | .801 | -1039.88 | 47.63 |

AIC = Akaike Informaton Criterion; BIC = Bayesian information criterion; SABIC = Sample adjusted BIC; LMR_LRT = Lo-Mendell-Rubin test; BLRT = bootstrap likelihood ratio test.

**p < .01

***p < .001.

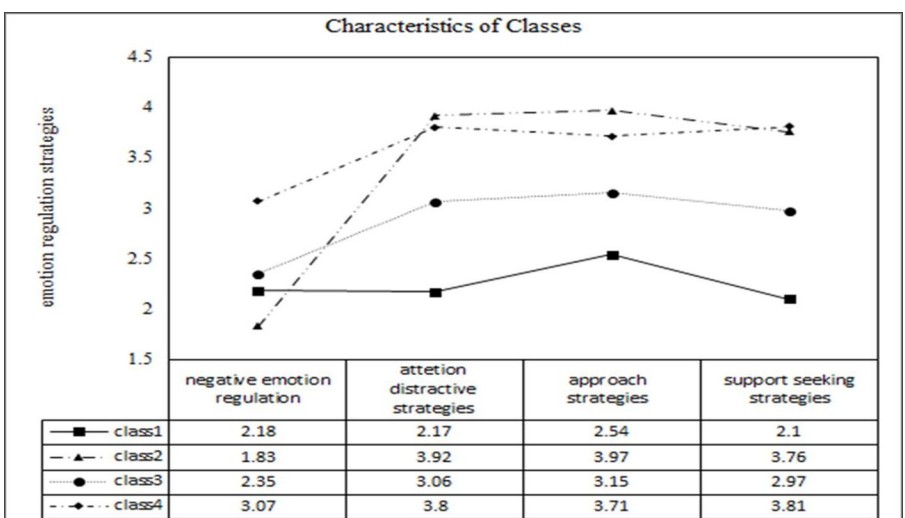

**Fig 2. Characteristics of classes.**

**Demographic and psychiatric symptoms predictors of emotional regulation classes.** In the study, the probability of each group being classified according to the age, job search period, and emotional psychiatric symptoms(depression, somatization, anxiety) of unemployed people was examined (see Table 3). First, the reference group of class 1(low emotion regulation class) showed that the longer the period of unemployment and the more the experience of somatization symptoms and anxiety, the higher the probability of belonging to class 1(low emotion regulation class) than class 2(adaptive emotion regulation class), and the higher the depression level and lower anxiety, the higher the probability of belonging to class 4(low emotion regulation class) than class 1(high negative emotion regulation-support seeking class). The result of comparing class 1(low emotion regulation class) and class 2(adaptive emotion regulation class) means that the longer the job search period and the higher the level of somatization and anxiety, the less positive emotion regulation strategies such as attention distractive, approach, and support seeking are used. Also, the result of comparing class 1(low emotion regulation class) and class 3(low negative emotion regulation-moderate positive regulation class) is similar, and the lower the somatization level, the more adaptive strategies are included in the group. Next, compared to class 2(adaptive emotion regulation class), class 3(low negative emotion regulation-moderate positive regulation class) showed higher levels of somatization and

**Table 3. Associations between psychosocial predictors and class membership.**

| | reference class 1 vs | | | | | | reference class 2 vs | | | | reference class 3 vs | |
|---|---|---|---|---|---|---|---|---|---|---|---|---|
| | 2 | | 3 | | 4 | | 3 | | 4 | | 4 | |
| | β | SE | β | SE | β | SE | β | SE | β | SE | β | SE |
| age | -1.38 | .05 | -0.39 | .03 | -0.34 | .03 | 1.30 | .05 | 1.26 | .05 | 0.14 | .03 |
| job seeking months | -2.58** | .02 | -0.27 | .01 | -0.13 | .01 | 2.59** | .02 | 2.77** | .02 | 0.16 | .01 |
| depression | 1.23 | .64 | 0.23 | .32 | 2.93** | .43 | -1.21 | .59 | 0.73 | .66 | 3.66*** | .32 |
| somatization | -2.90** | 1.93 | -2.13* | .73 | -1.80 | .66 | 2.28* | 1.78 | 2.34* | 1.89 | 0.58 | .61 |
| anxiety | -2.31* | 1.28 | -0.59 | 1.28 | -2.09* | .37 | 2.15* | 1.23 | 1.33 | 1.17 | -1.85 | .58 |

* $p < .05$

** $p < .01$.

anxiety, while class 4(high negative emotion regulation-support seeking class) showed higher levels of somatization. This means that unemployed people with high somatization and anxiety use less adaptive emotion regulation strategies. Both groups had longer job search months than class 2. The higher the level of depression, the higher the probability of belonging to class 4 (high negative emotion regulation-support seeking class) than class 3(low negative emotion regulation-moderate positive regulation class). This means that the group who has long unemployment and experienced depression symptoms is more likely to use maladaptive strategies and support seeking strategy that other adaptive regulation strategy.

## Discussion

This study explored the subgroups of people using emotion regulation strategies among the unemployed people and examined how psychiatric symptoms affected the probability of being classified into a group. As a results of study, four sub-classes(low emotion regulation class, adaptive emotional regulation class, low negative emotional regulation-moderate positive regulation class, high negative emotional regulation-support seeking class) were derived. In addition, it was found that in some classes, the probability of belonging to certain groups was affected by psychiatric symptoms.

Class 1(low emotion regulation class) was the lowest among the four groups in the use of adaptive emotion regulation strategies, and the use of negative emotion regulation strategies was also low. The results of this study are partly consistent with the results of Lee, Kim, and Choi [23] who observed low scores in overall emotional coping strategies. Lee et al. [23], found that the group with emotional coping levels was not significantly higher than the other groups, and the researchers interpreted this finding to be due to the fact that the level of emotional coping was low but other coping levels (e.g., problem-focused coping) were high, they would not experience maladjustment. In addition, it is consistent with the results of previous studies [6, 67] conducted in Korea that have shown that unemployed people are likely attempt to get out of a job-loss situation by immersing themselves in education or job search activities for reemployment after unemployment. In other words, it can be assumed that some unemployed people will use the coping strategy of problem-solving to gain reemployment rather than strategies to control their emotions or alleviate symptoms.

Class 2 was the adaptive emotion regulation class, which showed the lowest use of negative emotion regulation strategies and the highest use of positive emotion regulation strategies. The results of this study are consistent with the results of a previous study [23] that derived a "high-regulation group" that attempts to control emotion. Class 2 showed the highest average in the order of attentional distraction, approach, and support seeking in using adaptive emotion regulation strategies, and when compared with other classes, it was generally found that all three types of adaptive strategies were used frequently by the class. This indicates that there may be groups that use various adaptive emotion regulation strategies simultaneously. Therefore, in counseling, it is necessary to understand what the emotion regulation strategies are that the clients mainly use and train them to use various adaptive strategies according to the situation. For example, in situations where support-seeking strategies are difficult to use, counselors can teach clients how to distract attention from uncomfortable emotions or how to deal with emotions more actively.

Class 3 was the low negative emotion regulation-moderate positive regulation class, and all the emotion regulation strategies were used at moderate levels. This class uses various strategies (e.g., support seeking, attention distraction, and approach strategies) to positively change emotions. Similarly, Park and Kim [68] found that unemployed people used self-regulation strategies such as efforts to change their mood, focus on reality, tolerance or acceptance of

negative emotions, and efforts to stabilize emotions. The present study showed that, in this class, adaptive emotion regulation strategies are used evenly and involve individual adaptive efforts to control negative emotions. This class most reflects the cultural characteristics of Korea. In the case of the Korean people, they recognize their efforts as very important factors when coping with stress situations. Those who are unemployed try to overcome their difficulties by overcoming the related psychological difficulties.

Class 4 was the negative emotion regulation-support seeking class, which was the highest in the use of negative emotion regulation and the group that used support-seeking strategies most frequently among the adaptive emotion regulation strategies. The results of this study are partly consistent with the results of Lee et al. [23] who derived the "high divergence behavior group" with high negative behavior. Class 3 group is a group that blames others to control negative emotions in the unemployed situation or emits emotions in a safe situation, and uses negative methods such as rumination and addictive behavior.

As a result of examining the probability of being classified into each group according to demographic characteristics and psychological symptoms, the higher the somatization level, the higher the probability of being classified into class 1 (low emotion regulation class) than class 3(low negative emotion regulation-moderate positive regulation class). It means that the level of somatization is high and it is difficult to use support seeking, attention dispersion, and approach strategy. The results of this study are similar to those of a previous study [69] that found that people with high somatization symptoms are too careful to notice, amplify, perceive, or be caught up in symptoms and interpret the problem as physical causes. Lee, Lee, Yoon, Yang, Mun, Jung, and Eun [70] reported that patients who experienced somatization symptoms may have lower the use of the overall emotion regulation strategy because it attributes everything to the problem of the body rather than the attempt to control the emotion.

Class 2(adaptive emotion regulation class) was a group with a high probability of belonging to people with low psychological symptoms. Specifically, when compared with class 3(low negative emotion regulation class), the lower the level of anxiety, the higher the probability of belonging to class 2. This suggests that the higher the level of anxiety, the less the probability of using adaptive emotion regulation strategies and the less the attempt to recognize or control emotions [36].

Class 3 (low negative emotion regulation-moderate positive regulation class) was lower in depression level than class 4 (negative emotion regulation-support seeking class). In other words, depressed people use maladaptive strategies more frequently and prefer help-seeking strategies rather than emotional regulation strategies such as attentional dispersion or approach. These results are consistent with the results of previous studies [17, 20] that showed that depressed people use more negative emotional regulation strategies such as rumination and use limited active strategies such as problem solving and cognitive reconstruction.

The theoretical implications of this study are as follows. First, this study divided emotion regulation strategies into maladaptive strategy, attentional distraction strategy, approach strategy, and support-seeking strategy and examined which subgroup was formed according to emotion regulation strategies. This is meaningful in that it extended existing studies on emotion regulation strategies through a person-centered approach, assuming that everyone can use various emotion regulation strategies beyond the existing variable-centered approach that examined the relationship between the subtypes of emotion regulation strategies and psychiatric symptoms. In particular, it is meaningful that people mainly use some types of emotion regulation strategies and look at the characteristics of each type. Second, there have been claims that psychiatric symptoms is connected to the problem of emotion regulation strategies, and although emotion regulation related studies are actively conducted, there are limited studies on the relationship between various mental disorders and emotion regulation strategies.

Through this study, we examined the possibility of people with psychiatric symptoms being classified into types according to the use of emotion regulation strategies, and it is expected that it will be possible to understand emotion regulation strategies of people with psychiatric symptoms in counseling and treatment and to intervene in efficiently controlling emotions.

This study has the following practical implications: First, it was found that unemployed people can experience various psychological difficulties such as depression, anxiety, and somatization after job loss, and they may employ different emotion regulation strategies depending on the types of psychiatric symptoms. Recently, emotion regulation studies [61, 71] have mentioned that various emotion regulation strategies need to be used in accordance with context or situation, and negative strategies may have a positive effect in specific situations. Therefore, it is necessary to help the unemployed to have complementary emotion regulation strategies by exploring how they regulate their emotions according to psychological difficulties and how these emotion regulation strategies can be linked to the difficulties or long-term unemployment that they experience after the unemployment. For example, unemployed people with high depression often use negative emotion regulation strategies, whereas only support-seeking strategies among positive emotion regulation strategies are likely to be classified as high groups. Therefore, it is necessary for unemployed people with high depression to lower negative emotion regulation strategies and to increase various emotion regulation strategies, such as attentional distraction and approach emotion regulation strategy.

Second, in this study, group with relatively high anxiety showed less use of emotion regulation strategies than adaptive group. In other words, unemployed people with high anxiety are more likely to engage in problem-solving behavior, such as constantly trying to find a job or receiving necessary re-education rather than recognizing and solving their emotions. In counseling, it is necessary to examine the health of the group from a long-term perspective, and if necessary to lower the level of anxiety and to intervene in using adaptive emotion regulation strategies.

Finally, in the case of the most adaptive group, group 2, negative emotion regulation strategies were used the least, whiles approach strategy, attention distractive strategy, and support-seeking strategy were used evenly. Moreover, this study showed that when the use of negative emotion regulation strategies of unemployed people is similarly low, the lower the somatization and anxiety, the more likely they are to be classified as a group that uses positive emotion regulation strategies. This means that positive emotional regulation strategies can have a positive effect on somatization and anxiety regardless of the level of negative emotional regulation strategies. The results of this study are somewhat different from the results of previous studies [49] which showed significant relationship between positive emotional regulation and psychological problems only when negative emotional regulation strategies are high. Rather, This means that positive emotional regulation strategies are likely to be linked to specific psychiatric symptoms such as negative emotional regulation strategies. In counseling, it is necessary to teach the client how to use the most efficient emotion regulation strategies according to the situation of the client rather than teaching only one of these emotion regulation strategies. Specifically, it is necessary for unemployed people to recognize their emotions, use the most appropriate emotion regulation strategy for the situation, and develop the emotion regulation ability to objectively evaluate the reduction of negative emotions.

Despite these implications, this study has the following limitations. First, the unemployed people who participated in this study may have had more desire to find a job compared with others because they participated in training programs such as the Women's Job Center and the Vocational Education Center under the Ministry of Labor. In previous studies [46, 72], the presence of psychological symptoms was significantly different according to job search status and duration of unemployment. Second, since this study was a study conducted in Korea, it is

necessary to consider the cultural background when interpreting the results. In Korea, the fear and helplessness experienced during the financial crisis of 1997 had lasting effects, and the psychological difficulties experienced by heads of households who cannot provide for their families economically can be severe because of the patriarchal nature of the culture. In addition, because a long-term social security system for unemployed people is not yet established and fragmentary support is being provided, the psychological burden on the unemployed in Korea is higher than that of the unemployed in countries where the social security system is better established. Next, because the definition and measurement tools of emotion regulation strategies change with the researcher, it is necessary to considering the concepts and subtypes of emotion regulation strategies used in this study when interpreting the results. In particular, this study measured negative emotion regulation strategies as one factor based on Lee & Kwon's [73] findings, but it includes various types of negative emotion regulation strategies, such as blaming others, emotional expression in a safe situation, and addictive behavior. These strategies were classified as negative emotion regulation strategies through empirical research, but for unemployed people, there may be strategies with positive effects in the short term. Therefore, future studies should examine which groups are derived when more diverse classification criteria are used, and negative emotion regulation is treated as more than one factor. Finally, follow-up studies will need to use scales to measure more diverse mental health and behavioral problems. For example, according to age group, behavior problems such as Internet addiction may occur more frequently in young adults [74–76]. Therefore, it is necessary to examine whether each group according to the emotional regulation strategies of unemployed people has different behavioral results such as internet addiction and eating disorders.

## Author Contributions

**Conceptualization:** Min Sun Kim.

**Formal analysis:** Min Sun Kim.

**Methodology:** Min Sun Kim.

**Resources:** Min Sun Kim.

**Writing – original draft:** Min Sun Kim.

**Writing – review & editing:** Min Sun Kim.

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
