## [Decision Letter · Decision Letter 0]

19 Mar 2020

PONE-D-20-03549

Psychological symptoms and emotion regulation strategies among the unemployed person in Korea:

a latent profile analysis

PLOS ONE

Dear Dr. Kim,

Thank you for submitting your manuscript to PLOS ONE. After careful consideration, we feel that it has merit but does not fully meet PLOS ONE’s publication criteria as it currently stands. Therefore, we invite you to submit a revised version of the manuscript that addresses the points raised during the review process.

We would appreciate receiving your revised manuscript by May 03 2020 11:59PM. To enhance the reproducibility of your results, we recommend that if applicable you deposit your laboratory protocols in protocols.io, where a protocol can be assigned its own identifier (DOI) such that it can be cited independently in the future. For instructions see: http://journals.plos.org/plosone/s/submission-guidelines#loc-laboratory-protocols

We look forward to receiving your revised manuscript.

Kind regards,

Silvia Cimino

Academic Editor

PLOS ONE

Journal Requirements:

Reviewers' comments:

Reviewer's Responses to Questions

**Comments to the Author**

1. Is the manuscript technically sound, and do the data support the conclusions?

Reviewer #1: Partly

Reviewer #2: Yes

2. Has the statistical analysis been performed appropriately and rigorously? 

Reviewer #1: Yes

Reviewer #2: Yes

3. Have the authors made all data underlying the findings in their manuscript fully available?

Reviewer #1: Yes

Reviewer #2: Yes

4. Is the manuscript presented in an intelligible fashion and written in standard English?

Reviewer #1: Yes

Reviewer #2: Yes

5. Review Comments to the Author

Reviewer #1: Thank you very much for the possibility to review the manuscript titled “Psychological symptoms and emotion regulation strategies among the unemployed person in Korea: a latent profile analysis”. The paper is well written and the results are clearly supported by a robust methodology. I read the article with interest and I think it can add to previous literature and be published in this Journal, but it needs some revisions. Please find below some comments.

INTRODUCTION

The introduction is very rich and examines the results of the previous literature for the constructs under examination. However, the introduction is structured in a somewhat confusing way and does not guide the reader through the topic. In particular, after discussing the phenomenon of unemployment from a social point of view, highlighting the prevalence rates in Korea, the authors discuss suicide rates and then go on to discuss the effects on the whole household.

In this respect, before going into the specifics of suicide rates, it would be useful to introduce the fact that unemployment has a major impact on mental health and quality of life in general, and then go into the specifics. In particular, the study aimed to examine the strategies of emotional regulation, in a group of unemployed young people, and how these are associated with a series of psychopathological symptoms (in particular, depression, somatization, anxiety, and anger). However, In the description of the previous literature, the authors often discuss general studies on the relationship between specific constructs and then move on to specific studies on unemployed populations, and so on.

Often, for each of these paragraphs, it is specified what the present study intends to verify in several places in the text. Although the introduction goes into the various constructs adequately, with this organization the reader may feel confused. I suggest that we reorganize the introduction by initially reporting the studies that have shown significant associations between unemployment status and the psychopathological symptoms under examination. Then, discuss the results of the studies that analyzed the associations between psychopathological symptoms and emotional regulation strategies, and then move on to the description of the field studies specific to the unemployed population. Therefore, only at the end (in the subsection of The Present study) resuming the aims of the study, based on the previous literature which should be cited again to support the hypotheses of the study.

Furthermore, given that the population under examination are young adults, it would be useful to specify the role played by family support and the evolutionary tasks related to the construction of identity and autonomy in this specific evolutionary phase and from which theoretical perspective. Also, among the negative effects associated with unemployment, recent studies have highlighted the presence of an increased risk of developing Internet Addiction, which in turn other studies have shown may be a strategy to cope with psychological sufferance.

In this regard, see and cite the studies by Kim et al. (2017). Poor sleep quality and suicide attempt among adults with internet addiction: A nationwide community sample of Korea. PloS one, 12(4); Ballarotto et al. (2018). Adolescent Internet abuse: A study on the role of attachment to parents and peers in a large community sample. BioMed research international, 2018; Cimino, S., & Cerniglia, L. (2018). A longitudinal study for the empirical validation of an etiopathogenetic model of internet addiction in adolescence based on early emotion regulation. BioMed research international, 2018.

Furthermore, recent literature has shown that the relationship between psychopathological symptoms and emotional regulation strategies is complex and bidirectional. See, for example, the study by Schäfer et al. (2017). Emotion regulation strategies in depressive and anxiety symptoms in youth: A meta-analytic review. Journal of youth and adolescence, 46(2), 261-276; the study by Aldao, A., & Nolen-Hoeksema, S. (2012). When are adaptive strategies most predictive of psychopathology?. Journal of abnormal psychology, 121(1), 276; and the study by Porreca, et al., (2018). Emotional availability, neuropsychological functioning, and psychopathology: The context of parental substance use disorder. BioMed research international, 2018.

Furthermore, in some points, I suggest to clarify more clearly what the previous literature said. For example, authors write that “Previous studies[38, 41] have investigated the relationship between emotion regulation strategies and negative emotions, emotional disorders, and psychopathology”, describing the regulatory strategies that have emerged but do not report the results related to the associations that have emerged between these constructs.

In other points, the authors refer to previous studies, but without mentioning them. For example, "First, the research on unemployed people is still focused on examining the relationship between

emotion regulation strategies and psychological symptoms, and the number of studies is very

limited." What studies? What did they find?

In many places in the text, the authors refer to "previous studies", "some studies", but without explicitly citing them. These should be cited every time they are mentioned. In the same way, I suggest citing bibliographic sources referring to the difference between the variable-centered approach and person-centered approach.

Moreover, since the sample is composed of predominantly female subjects, it is important to highlight gender-related differences in psychopathological symptomatology and emotional regulation strategies.

The last line of the introduction ( “The model of this study is as follows”) is written with a font size smaller than the rest of the text.

METHODS AND DISCUSSION

In the subsection “Participants and procedure” the authors wrote the percentage of participants who had more than 2 years of jobless experience was high. what is this percentage?

What were the criteria for inclusion and exclusion of subjects?

In the last line of this subsection, the authors wrote in red that they received a

consent form for all participants. Why in red? In which form did you obtain consent? Written? Oral?

The emotion regulation strategy questionnaire(ERSQ) is a self-report? Is there a Likert scale? In the description of the instrument authors wrote that “Maladaptive strategies showed significant positive correlation with psychopathology such as depression, while other strategies showed negative correlation” and that “This study classified latent profiles according to emotional regulation strategies of unemployed people based on ERSQ scale, which measures various emotional regulation strategies”. I suggest to delete these parts or move them respectively in the introductions (citing the literature) and in the description of the procedure, but not in the description of the instrument. Moreover, the authors did not report the reliability of ERSQ scales for this study.

In Table 1 there are some typing errors: “attetion distractive strategies” should be “attention”; “state angger” and “ trait angger” should be anger.

Among the socio-demographic variables (age, job seeking Months) why did you not consider the role played by the gender, considering that 75.9% of the sample is female? I suggest to do it.

As suggested above, Also in the discussions, it is important to report the studies of recent literature that have shown a bidirectional effect between psychopathological symptoms and emotional regulation strategies. In addition, in many points of the discussions there is a lack of sources in the previous literature on which the discussions of some results are based, for example, “This is consistent with the results of previous studies”, “Unlike previous studies”, “This is in line with findings from previous studies” etc. What previous studies? They should always be cited.

Reviewer #2: Thank you very much for the possibility to review the article titled "Psychological symptoms and emotion regulation strategies among the unemployed person in Korea: a latent profile analysis". This study is very interesting and I believe that it can be published if the authors make few revisions.

Specifically, the Introduction is very broad and it would be useful that in the section "The present study", the authors explain more clearly the hypotheses underlying the present study, in line with the literature.

Furthermore, there are many typing errors in the text (such as at the beginning of page 9 or on page 12 "the emotionr egulation", etc.). Authors are invited to completely revise the text.

In addition, the numbering of references should follow their appearance in the text. Authors are therefore invited to completely revise the numbering of references.

6. PLOS authors have the option to publish the peer review history of their article (what does this mean?). If published, this will include your full peer review and any attached files.

Reviewer #1: No

Reviewer #2: No

---

## [Author Response · Author response to Decision Letter 0]

26 Jun 2020

May 30, 2020

Dear Dr. Silvia Cimino:

Thank you for giving us an opportunity to revise and resubmit our manuscript entitled “Psychiatric symptoms and emotion regulation strategies among the unemployed people in Korea: a latent profile analysis” (Manuscript number: PONE-D-20-03549). I am Min Sun Kim, and I will be serving as the first author and corresponding author for this version of the manuscript. 

Below I will describe how we have addressed the issues you raised concerning the previous version of the manuscript and also how the revision has implemented the reviewers’ suggestions. I paste in your and the reviewers’ comments and describe how we have addressed each one. To make the changes easier to identify, where necessary, we have numbered them. 

I’d like to thank you again for this opportunity and look forward to hearing from you soon. 

Sincerely,

Min Sun Kim, Ph.D.

Reviewer1:

INTRODUCTION

1. The introduction is very rich and examines the results of the previous literature for the constructs under examination. However, the introduction is structured in a somewhat confusing way and does not guide the reader through the topic. In particular, after discussing the phenomenon of unemployment from a social point of view, highlighting the prevalence rates in Korea, the authors discuss suicide rates and then go on to discuss the effects on the whole household.

In this respect, before going into the specifics of suicide rates, it would be useful to introduce the fact that unemployment has a major impact on mental health and quality of life in general, and then go into the specifics. 

: R1, in response to your editorial comments, the introduction has been modified on pp. 3-4. 

 In the absence of a social security system, unemployment not only poses a fundamental threat to the right to live for individual workers, but also has a negative effect on the entire family, which can threaten the structure of the family and the overall social structure. The prevalence of family crises among Koreans is 50%, and the frequency of economic crisis (e.g., unemployment) was the highest among the family crises[35]. In periods of economic recession and overall industrial change, problems related to unemployment can be a factor that makes the social cost more burdensome. Therefore, it is necessary to expand mental health programs that can facilitate the process from unemployment to reemployment, and to focus more on the mental health problems of unemployed people at the national level. 

 From a psychological perspective, unemployment is linked to the following negative results: First, it was found that the loss of economic independence and social role performances due to the decrease of economic resources obtained from a fixed income were insolvent. Second, it leads to involuntary interruption of the mutual relationship between individuals and the organization. Third, unemployment means loss of work, which, for many, is the basis of achievement, self-realization, and self-identity[19, 34]. Also, the increase in anxiety and tension due to unemployment has been demonstrated by many empirical studies [16, 31]. McKee-Ryan, Song, Wanberg, and Kinicki [53] examined 52 cross-sectional studies that compared the health of unemployed and employed people and found that those who were unemployed had significantly poorer mental health, life satisfaction, family satisfaction, and subjective physical health. Unemployment has been linked to increases in the suicide rate among middle-aged people. According to the distribution trend of suicide rates by age in Korea (the number of suicides per 100,000 ages), the suicide rate of young people in their 20s was the highest in 1983, but the suicide rate of young people in their 20s was lowered in 1993, while the suicide rate of older people rose[39]. Comparing the distribution of suicide rates by age in 1993 and 2003, it can be seen that the suicide rate is increasing as the age of the population increases, and the suicide rate of adults aged over 50 years has been increasing very rapidly recently. 

2. In particular, the study aimedto examine the strategies of emotional regulation, in a group of unemployed young people, and how these are associated with a series of psychopathological symptoms (in particular, depression, somatization, anxiety, and anger). However, In the description of the previous literature, the authors often discuss general studies on the relationship between specific constructs and then move on to specific studies on unemployed populations, and so on.

: Based on the first revision, we recruited additional unemployed groups of all ages, not unemployed young people. In the previous study, I wanted to add a study on the unemployed group, but there were limitations in the studies that looked at the relationship between emotional regulation strategies and psychiatric symptoms.

3. Often, for each of these paragraphs, it is specified what the present study intends to verify in several places in the text. Although the introduction goes into the various constructs adequately, with this organization the reader may feel confused. I suggest that we reorganize the introduction by initially reporting the studies that have shown significant associations between unemployment status and the psychopathological symptoms under examination. Then, discuss the results of the studies that analyzed the associations betweenpsychopathological symptoms and emotional regulation strategies, and then move on to the description of the field studies specific to the unemployed population. Therefore, only at the end (in the subsection of The Present study) resuming the aims of the study, based on the previous literature which should be cited again to support the hypotheses of the study.

: In response to your comments, we have refined the structure of this paper and added research and research questions to support the hypothesis of this study in the present study as below.

(pp. 9-10)

In this study, we have focused on identifying the characteristics of each subgroup by classifying unemployed poeple into subgroups according to the use of emotional regulation strategies through latent profile analysis. This study derived latent profiles based on one negative emotion regulation strategy and four adaptive emotion regulation strategies (Fig 1). We also examined whether the probability of belonging to each group derived based on emotion regulation strategies differs according to the psychiatric symptoms of unemployed people. According to previous studies, people with vulnerability to depression were more likely to use negative emotional regulation strategies such as avoidance[23], luminance[23, 56], and reinterpretation[25]. In addition, the group with low emotional acceptance and awareness showed higher anxiety level[10, 64], and the group with low emotional acceptance and awareness showed higher somatization in the group without expressing negative emotions[60, 74]. In this study, the probability of being classified into each group would be different according to psychiatric symptoms of the unemployed people.

Research Question 1: Are there qualitatively distinct emotional regulation strategy profile of unemployment people?

Research Question 2: Do demographic information and psychiatric symptoms predict emotional regulation strategy profile membership?

4. Furthermore, given that the population under examination are young adults, it would be useful to specify the role played by family support and the evolutionary tasks related to the construction of identity and autonomy in this specific evolutionary phase and from which theoretical perspective. Also, among the negative effects associated with unemployment, recent studies have highlighted the presence of an increased risk of developing Internet Addiction, which in turn other studies have shown may be a strategy to cope with psychological sufferance.

In this regard, see and cite the studies by Kim et al.(2017). Poor sleep quality and suicide attempt among adults with internet addiction: A nationwide community sample of Korea. PloS one, 12(4); Ballarotto et al. (2018). Adolescent Internet abuse: A study on the role of attachment to parents and peers in a large community sample. BioMed research international, 2018; Cimino, S., & Cerniglia, L. (2018). A longitudinal study for the empirical validation of an etiopathogenetic model of internet addiction in adolescence based on early emotion regulation. BioMed research international, 2018.

: Thank you for your good point. This study included participants of various age groups.

Therefore, it is not a group directly related to problems such as Internet addiction and it is not included in the part.

5. Furthermore, recent literature has shown that the relationship between psychopathological symptoms and emotional regulation strategies is complex and bidirectional. See, for example, the study by Schäfer et al. (2017). Emotion regulation strategies in depressive and anxiety symptoms in youth: A meta-analytic review. Journal of youth and adolescence, 46(2), 261-276; the study by Aldao, A., & Nolen-Hoeksema, S. (2012). Whenare adaptive strategies most predictive of psychopathology?. Journal of abnormal psychology, 121(1), 276; and the study by Porreca, et al., (2018). Emotional availability, neuropsychological functioning, and psychopathology: The context of parental substance use disorder. BioMed research international, 2018.

: After reviewing the references you recommended, we added the contents to the paper as below.

(pp. 8-9) However, recent studies have shown that there are some inconsistent results in explaining the relationship between psychopathology, positive emotional regulation strategies, and negative emotional regulation strategies. For example, Schfer, Naumann, Holmes, Tuschen-Caffier, and Samson[61], which meta-analyzed existing studies on the relationship between emotional regulation strategies, depression, and anxiety, have shown that positive emotional regulation strategies such as cognitive reinterpretation have more important effects on psychiatric symptoms such as depression and anxiety than negative emotional regulation strategies such as rumination and avoidance. This means that enhancing positive emotional regulation strategies can have a more positive effect on depression and anxiety of adolescents than reducing negative emotional regulation. On the other hand, Aldao and Nolen[3]'s study showed that the relationship between positive emotion regulation strategies and psychiatric symptoms were more significant for adults who use high negative emotion regulation strategies. This means that the positive emotion regulation strategies is more effective in lowering the psychopathology of people who use negative emotion regulation and people use negative and positive emotion regulation strategies flexibly according to the situation (e.g., 31, 75). 

(p. 24) Finally, in the case of the most adaptive group, group 2, negative emotion regulation strategies were used the least, whiles approach strategy, attention distractive strategy, and support-seeking strategy were used evenly. Moreover, this study showed that when the use of negative emotion regulation strategies of unemployed people is similarly low, the lower the somatization and anxiety, the more likely they are to be classified as a group that uses positive emotion regulation strategies. This means that positive emotional regulation strategies can have a positive effect on somatization and anxiety regardless of the level of negative emotional regulation strategies. The results of this study are somewhat different from the results of previous studies[3] which showed significant relationship between positive emotional regulation and psychological problems only when negative emotional regulation strategies are high. Rather, This means that positive emotional regulation strategies are likely to be linked to specific psychiatric symptoms such as negative emotional regulation strategies. In counseling, it is necessary to teach the client how to use the most efficient emotion regulation strategies according to the situation of the client rather than teaching only one of these emotion regulation strategies. Specifically, it is necessary for unemployed people to recognize their emotions, use the most appropriate emotion regulation strategy for the situation, and develop the emotion regulation ability to objectively evaluate the reduction of negative emotions.

6. Furthermore, in some points, I suggest to clarify more clearly what the previous literature said. For example, authors write that “Previous studies[38, 41] have investigated the relationship between emotion regulation strategies and negative emotions, emotional disorders, and psychopathology”, describing the regulatory strategies that have emerged but do not report the results related to the associations that have emerged between these constructs. In other points, the authors refer to previous studies, but without mentioning them. For example, "First, the research on unemployed people is still focused on examining the relationshipbetween

emotion regulation strategies and psychological symptoms, and the number of studies is very

limited." What studies? What did they find?

: In response to your comments, we deleted the unnecessary part and described the study clearly.

7. In many places in the text, the authors refer to "previous studies", "some studies", but without explicitly citing them. These should be cited every time they are mentioned. In the same way, I suggest citing bibliographic sources referring to the difference between the variable-centered approach and person-centered approach.

Moreover, since the sample is composed of predominantly female subjects, it is important to highlight gender-related differences in psychopathological symptomatology and emotional regulation strategies.

: In response to your comments, we deleted those expression and re-write the sentence. 

The last line of the introduction ( “The model of this study is as follows”) is written with a font size smaller than the rest of the text.

: (p. 10) In response to your comments, we have revised that sentence.

8. In the subsection “Participants and procedure” theauthors wrote the percentage of participants who had more than 2 years of jobless experience was high. what is this percentage?

: R1, in response to your editorial comments, that sentence has been modified.

(p. 11) The gender of the participants was 136 males(42.8%) and 182 females(57.2%), and the averageage was 42.55(SD=7.40). The educational background was 70(22.0%) high school graduates, 37(11.6) technical college graduates, 182(57.2%) university graduates, and 29(9.1%) graduate students. The retirement types were voluntary retirees(58.2%) and involuntary retirees(41.8%). The number of jobless was 55(17.3%) once, 103(32.4%) twice, and 160(50.3%) three times and more. The average job search period was 35.84(SD=26.83) months. 

9. What were the criteria for inclusion and exclusion of subjects?

: In response to your editorial comments, we added the criteria for selecting participants.

(p. 11) In order to recruit participants, the purpose of the study was explained to the institutions that help reemployment or job search such as the vocational education center under the Ministry of Labor and the Women's Job Center. In this study, participants were selected from adults over 20 years old who do not currently have a full-time job and want to find a job. 

10. In the last line of this subsection, the authors wrote in red that they received a

consent form for all participants. Why in red? In which form did you obtain consent? Written? Oral?

: This study explained the purpose of the study and received oral consent. The contents were added as follows.

(p. 11) In order to recruit participants, the purpose of the study was explained to the institutions that help reemployment or job search such as the vocational education center under the Ministry of Labor and the Women's Job Center. In this study, participants were selected from adults over 20 years old who do not currently have a full-time job and want to find a job. The study was conducted in accordance with the Declaration of Helsinki. The survey kits were distributed to the institutions that allowed the participation in the study, and the questionnaires were distributed to the people who were willing to participate in the study. This study explained the necessity and purpose of the study to the participants and received oral consent. The researchers explained that there is no disadvantage due to the absence of research and that they can withdraw if there is no intention to participate in the survey. 

11. The emotion regulation strategy questionnaire(ERSQ) is a self-report? Is there a Likert scale? In the description of the instrument authors wrote that “Maladaptive strategies showed significant positive correlation with psychopathology such as depression, while other strategies showed negative correlation” and that “This study classified latent profiles according to emotional regulation strategies of unemployed people based on ERSQ scale, which measures various emotional regulation strategies”. I suggest to delete these parts or move them respectivelyin the introductions (citing the literature) and in the description of the procedure, but not in the description of the instrument. Moreover, the authors did not report the reliability of ERSQ scales for this study.

: We thought it would be appropriate to include the explanation of the scale in the scale part, so we summarized the contents of the theoretical background and presented it together. We also added more detailed description of the scale.

(p. 12) In this study, the emotion regulation strategy questionnaire(ERSQ) developed by Lee and Kwon[45] was used to measure the emotion regulation strategy. ERSQ is divided into cognitive, experiential, and behavioral strategies according to what aspect of emotion is approach ed and emotional changes are caused. A total of 16 strategies including 5 cognitive strategies, 5 experiential strategies, and 6 behavioral strategies are measured. A result of Lee and Kwon[45]'s research, 16 emotion regulation strategies were classified into maladaptive strategies, support-seeking strategies, attention distraction strategies, and approach strategies. Maladaptive strategies showed significant positive correlation with psychopathology such as depression, while other strategies showed negative correlation. Previous research classified the other three strategies as adaptive ones except for maladaptive emotion regulation[45]. Lee and Kwon referred to support-seeking strategies, attraction differentiation strategies, and approach strategies as adaptive strategies. This study classified latent profiles according to emotional regulation strategies of unemployed people based on ERSQ scale, which measures various emotional regulation strategies.

12. In Table 1 there are some typing errors: “attetion distractive strategies” should be “attention”; “state angger” and “ trait angger” should be anger.

: R1, in response to your editorial comments, we modified those words.

13. Among the socio-demographic variables (age, job seeking Months) why did you not consider the role played by the gender, considering that 75.9% of the sample is female? I suggest to do it.

: R1, in response to your editorial comments, we recruited additional participants to adjust the gender ratio.

(p. 11) The gender of the participants was 136 males(42.8%) and 182 females(57.2%), and the averageage was 42.55(SD=7.40). The educational background was 70(22.0%) high school graduates, 37(11.6) technical college graduates, 182(57.2%) university graduates, and 29(9.1%) graduate students. The retirement types were voluntary retirees(58.2%) and involuntary retirees(41.8%). The number of jobless was 55(17.3%) once, 103(32.4%) twice, and 160(50.3%) three times and more. The average job search period was 35.84(SD=26.83) months. 

14. As suggested above, Also in the discussions, it is important to report the studies of recent literature that have shown a bidirectional effect between psychopathological symptoms and emotional regulation strategies. 

: After reviewing the references you recommended, we added the contents to the paper as below. 

(p. 24) Finally, in the case of the most adaptive group, group 2, negative emotion regulation strategies were used the least, whiles approach strategy, attention distractive strategy, and support-seeking strategy were used evenly. Moreover, this study showed that when the use of negative emotion regulation strategies of unemployed people is similarly low, the lower the somatization and anxiety, the more likely they are to be classified as a group that uses positive emotion regulation strategies. This means that positive emotional regulation strategies can have a positive effect on somatization and anxiety regardless of the level of negative emotional regulation strategies. The results of this study are somewhat different from the results of previous studies[3] which showed significant relationship between positive emotional regulation and psychological problems only when negative emotional regulation strategies are high. Rather, This means that positive emotional regulation strategies are likely to be linked to specific psychiatric symptoms such as negative emotional regulation strategies. In counseling, it is necessary to teach the client how to use the most efficient emotion regulation strategies according to the situation of the client rather than teaching only one of these emotion regulation strategies. Specifically, it is necessary for unemployed people to recognize their emotions, use the most appropriate emotion regulation strategy for the situation, and develop the emotion regulation ability to objectively evaluate the reduction of negative emotions.

15. In addition, in many points of the discussions there is a lack of sources in the previous literature on which the discussions of some results are based, forexample, “This is consistent with the results of previous studies”, “Unlike previous studies”, “This is in line with findings from previous studies” etc. What previous studies? They should always be cited.

: In response to your comments, we added the references. 

Reviewer #2

1. Specifically, the Introduction is very broad and it would be useful that in the section "The present study", the authors explain more clearly the hypotheses underlying the present study, in line with the literature.

: In response to your comments, we have added research and research questions to support the hypothesis of this study in the present study as below.

(pp. 9-10)

 In this study, we have focused on identifying the characteristics of each subgroup by classifying unemployed poeple into subgroups according to the use of emotional regulation strategies through latent profile analysis. This study derived latent profiles based on one negative emotion regulation strategy and four adaptive emotion regulation strategies (Fig 1). We also examined whether the probability of belonging to each group derived based on emotion regulation strategies differs according to the psychiatric symptoms of unemployed people. According to previous studies, people with vulnerability to depression were more likely to use negative emotional regulation strategies such as avoidance[23], luminance[23, 56], and reinterpretation[25]. In addition, the group with low emotional acceptance and awareness showed higher anxiety level[10, 64], and the group with low emotional acceptance and awareness showed higher somatization in the group without expressing negative emotions[60, 74]. In this study, the probability of being classified into each group would be different according to psychiatric symptoms of the unemployed people.

Research Question 1: Are there qualitatively distinct emotional regulation strategy profile of unemployment people?

Research Question 2: Do demographic information and psychiatric symptoms predict emotional regulation strategy profile membership?

2. Furthermore, there are many typing errors in the text (such as at the beginning of page 9 or on page 12 "the emotionr egulation", etc.). Authors are invitedto completely revise the text.

: In response to your comments, we have revised those typing errors. 

3. In addition, the numbering of references should follow their appearance in the text. Authors are therefore invited to completely revise the numbering of references.

: We have modified the list of references.

---

## [Editor Report · Decision Letter 1]

17 Jul 2020

Psychiatric symptoms and emotion regulation strategies among the unemployed people in Korea: a latent profile analysis

PONE-D-20-03549R1

Dear Authors,

We’re pleased to inform you that your manuscript has been judged scientifically suitable for publication and will be formally accepted for publication once it meets all outstanding technical requirements.

Kind regards,

Silvia Cimino

Academic Editor

PLOS ONE

---

## [Editor Report · Acceptance letter]

21 Jul 2020

PONE-D-20-03549R1 

Psychiatric symptoms and emotion regulation strategies among the unemployed people in Korea: a latent profile analysis 

Dear Dr. Kim:

I'm pleased to inform you that your manuscript has been deemed suitable for publication in PLOS ONE. Congratulations! Your manuscript is now with our production department. 

Kind regards, 

on behalf of

Professor Silvia Cimino 

Academic Editor

PLOS ONE